# Do Countries Need Religious and Educational Freedoms to Achieve Prosperity?

**Khatib Ahmad Khan** [1,*] , **Danabekova Aigerim** [2] **and Yansheng Wu** [2]

1   School of Oriental Studies, Xi'an International Studies University, Xi'an 710128, China
2   School of Chinese Language and Literature, Shaanxi Normal University, Xi'an 710119, China
*   Correspondence: khatib786@snnu.edu.cn

**Abstract:** This study examines the impact of religious and educational freedoms on prosperity. The system GMM model is applied by using the data of 45 lower-, middle-, and high-income countries from 2009 to 2018. The results show that religious and academic freedoms are positively and statistically significantly associated with prosperity. It is revealed from the results that the lagged impact of both religious and education freedoms has a higher impact on prosperity than the current level of both variables. Interestingly, the interaction term between academic and religious freedom is also positive and statistically significant, indicating that their combined effect further increases prosperity. Further, the interaction term between government effectiveness and gross fixed capital formation is introduced. Its impact is positive and significant, indicating that capital investment positively affects prosperity in the case of higher government effectiveness. This study uses gross fixed capital formation and trade openness as control variables and these variables have a positive impact on prosperity, but the impact of trade openness on prosperity is insignificant. Thus, this study recommends religious and education freedom to achieve prosperity, especially in low-income countries that are already lagging.

**Keywords:** system GMM; religious freedom; educational freedom; development; prosperity





## 1. Introduction

Religious activities are increasing around the world (Thomas 2010). The religious revival is distinguished not only by the rise of fundamentalism and a fervent dedication to a certain set of rituals and beliefs but also by a wide range of restored rites and practices, both public and private (Macedo and Macedo 2009).

The globe is confronted with many difficult concerns regarding addressing issues such as expanding populations, rising poverty levels, global pandemics, and civil instability (Rasul et al. 2021; Qureshi 2021). As we enter the new century, it is critical to develop realistic economic models that employ elements that most support the circumstances required for prosperity to grow and to define the appropriate objectives that should be pursued to achieve such conditions (Robèrt 2000; Rodrik and Sabel 2020).

The world experienced a surge of religious intolerance in the early twenty-first century. Even a small number of people driven by religious extremism could start wars and cause significant economic disruptions, as evidenced by the terrorist attacks of 11 September 2001 (Mockaitis 2008; Kepel 2021). Aside from such battles, studies reveal that individuals currently living with high levels of religious hostility or anger will clash; a far larger number than only five years earlier (Hacker 2010; Kochhar and Fry 2014). An increasing tide of government limitations on religious freedom has accompanied this surge of religious hostility, bloodshed, and conflict. People living under severe government control climbed to 64% in 2012 from 58% of the total population (Saiya 2015; Rieffer-Flanagan 2022).

The increase in religious hostility and government limits function at the same time. Alon et al. (2017) found that there is a direct correlation between popular antipathy

toward religion and government disregard for religious freedom. Indeed, past theory and research go beyond proving correlations to establishing causality. According to the religious economics hypothesis, when government limits on religious freedom rise, negative results such as more anger towards religion and society arise (Grim and Finke 2007). This theory has experimentally proven by using multivariate tests that show government constraints on religious liberty are the largest reason for religious anger and clashes, even when other theoretical, economic, political, social, and demographic variables are controlled for (Astor et al. 2017; Fox 2015).

Religious hostility and prohibitions create cultures that may deter domestic and international investment, impede sustainable development, and destabilize whole economies. Such is the case in Egypt's unending cycle of religious rule and hostility, which has harmed the tourist industry (Gershoni and Jankowski 2009). A few recent instances from Muslim-majority nations with especially high degrees of religious restriction (Grim et al. 2014) demonstrate how religious freedom correlates to poor economic and commercial performance.

Religious restrictions take numerous forms in Muslim-majority nations. Islamic banking is one direct religious constraint that impacts economic freedom. The Islamic law board decides a particular Islamic financial instrument is acceptable or not on behalf of businesses who create, buy, or sell these instruments (Kobeissi 2005); these instruments' acceptance at stock exchange is often decided by differing interpretations of Islamic anti-blasphemy legislation used to target commercial competitors (Uddin and Tarin 2013). Perhaps most importantly for imminent economic development, the insecurity associated with high and escalating religious restrictions and hostility may persuade young entrepreneurs to choose other opportunities (Acemoglu et al. 2005)

Research has shown, in general, that freedom of religion, as measured by the nonexistence of fierce religious persecution and conflict, is an essential component of peace and stability (Grim et al. 2014). This is also very important for the business world, because when there is stability, there are more opportunities for investments as well as regular and predictable business operations, and this is especially true in markets that are expanding and becoming more competitive. Beyond preserving calm and order, the freedom to practice one's religion, as with freedom in general, may also contribute to the development of society for the better (Shilliam 2012). In order for a society to progress, Amartya Sen (1999, p. 3) asserts that the factors that contribute to "unfreedom" must be eradicated. According to Sen's line of reasoning, restrictions placed on religious practice contribute to a lack of freedom. The expansion of religious liberty also leads to the expansion of other kinds of liberty. According to the findings of various pieces of research, the presence of religious freedom is strongly associated with other autonomies, to the point where it can be measured as a hustled product of other autonomies that are strongly associated with a number of favorable social and economic outcomes, from better health care to higher wages for women (Grim et al. 2014). There is a correlation between the freedom of religion and the reduction of corruption, which is one of the most important factors in sustained economic growth. For instance, regulations and practices that make religion more difficult to practice are linked to higher levels of corruption (Lipset and Lenz 2000; Aqeel et al. 2022). In eight of the ten most corrupt countries in the world, religious freedom is subject to significant restrictions imposed by the state (Moaddel and Karabenick 2018).

The connection between liberty in religion and favorable economic outcomes may be better understood using different theoretical frameworks. Because it acknowledges that "religious participation is an economic activity and that religious freedom leads in greater religious involvement and consequently more economic development", the religious economy model may offer the most direct causal pathway. It recognizes that the reason for this is that "religious participation is an economic activity and that religious freedom leads to higher religious involvement." Given that religious groups are permitted to operate in a setting that is both free and competitive, religion may in fact provide discernible advantages for a society's human and social development (Gill 2013). For instance, Robert Woodberry (2012) found a link between the expansion of the world economy and the rise

in the proportion of Protestant churches and denominations that carry out missionary endeavors to win over new members. According to De Tocqueville (1955) Protestant associations established seminaries, inns, and churches, distributed books, and established hospitals, prisons, and schools in the USA during the nineteenth century in a vulnerable and generally free environment with other religious and civic relations.

The advantages of economic development are based on economic growth and competitiveness. To achieve economic growth, regions must foster innovation and research and development (R&D). Thus, human capital progress is a significant predictor of growth and development since (working) people are responsible for creativity and innovation. Individuals' hard work and dedication, as well as the competence and enthusiasm of instructors in a supportive atmosphere, produce talents with creative ideas. As a result, new ideas and start-ups often emerge as by-products of high-quality higher education institutions and tend to relocate into competitive and benchmark-enabling employment settings (Handler 2021).

In practically every nation on the planet, there is debate regarding the degree of religious and educational freedom. Around the world, there is an expansion in both religion and education, which has led to a more variable trajectory for religious and educational freedoms. Given the importance of education and religious freedom in low-, middle-, and high-income countries, which was mostly ignored in the previous study, this research examines the impact of educational and religious freedoms on prosperity. The system GMM model is proposed for the panel data. The findings demonstrate a favorable and statistically significant relationship between freedom of religion and freedom of education and prosperity.

Education may be seen of as a type of moral training in addition to its more common meanings as a means of acquiring knowledge and developing one's skill set. Undoubtedly, it is the sharing of facts and figures, but it is also the passing on of expertise, understanding, and insight. The passing of values and beliefs from one generation to the next is a major goal of formal education. A human right to education is the method by which one might gain freedom and become a real individuated creature, self-aware yet profoundly and sincerely linked to prosperity (Hodgson 1998).

This study is organized as follows: we start with the introduction, followed by the research method, which is discussed in Section 2. Section 3 is about results and discussion. Section 4 concludes the study.

## 2. Research Method

### 2.1. System GMM

There are two widely used techniques for analyzing the model setting of education and religious freedom's effects on prosperity: cross-sectional data regression and panel data construction. Cross-sectional regression has certain issues related to the endogeneity of variables and the static characteristics of individuals. The dynamic connection between the year's statistics cannot be expressed. In recent years, panel data have grown to a significant level. Panel data collect both transverse and time period-related information. Panel data represent both time and place; the data do not just sum up the data but instead employ the intercept item to reflect individual variations in the dynamic evolution of the data, hence lowering the estimate bias due to neglecting individual differences. Panel data features are included in the level of information, increasing the sample size and strengthening the estimate findings. Consequently, this work builds a model using panel data to perform research (Cao et al. 2020). The general dynamic panel model is

$$HDI_{it} = \alpha_i + \beta_1 HDI_{i.t-1} + \beta_2 Religfr_{it} + \beta_3 Religfr_{i,t-1} + \beta_4 Edufr_{it} + \beta_4 Edufr_{it-1} + \beta_5 GE_{it} + \beta_6 TO_{it} \atop + \beta_7 GFCF_{it} + \varepsilon_{it} \qquad c \qquad (1)$$

where *HDI* is the dependent variable, *Religfr* is religious freedom, *Edufr* is educational freedom, *GE* is government effectiveness, *TO* is trade openness, *GFCF* is gross fixed capital

formation, $\alpha_i$ an unobservable random variable representing individual heterogeneity, and $\varepsilon_{it}$ is a random disturbance term.

The preceding period's dependent variable $HDI_{i,t-1}$ is introduced into the equation as an independent variable by the dynamic panel data, which increases the model's realism by implying aspects that were not taken into account in the model. The random disturbance term and the explanatory variable, however, could be connected. The dynamic panel data model can achieve accurate and unbiased parameter estimates to address the aforementioned issues and prevent distortions brought on by biased parameters. Heteroscedasticity, autocorrelation, and individual effects are prevalent in the dynamic panel data model. In general, there are two techniques to assess it. One way to increase accuracy is to adjust the estimator that was derived from the general static model by reducing estimate error. The generalized moment estimation (GMM) approach is used to directly estimate the model. Many academics have taken an interest in GMM, since it can provide consistent estimate findings all at once. As a result, the system GMM model estimate approach model is chosen in this research. Blundell et al. (1995) and Arellano and Bover (1995) also put forward this estimating technique. To increase the effectiveness of parameter estimation, it combines the differential GMM and horizontal GMM estimation approaches. This article establishes the general form of the GMM system for quick explanation. All the variables were converted using a logarithmic scale in accordance with Barro and McCleary (2003) in order to eliminate any potential heteroscedasticity (i.e., an uneven distribution of the variance) and translate the regression coefficients into elasticity measurements.

$$lHDI_{it} = \alpha_i + \beta_1 lHDI_{i.t-1} + \beta_2 lReligfr_{it} + \beta_3 lReligfr_{i,t-1} + \beta_4 lEdufr_{it} + \beta_4 lEdufr_{it-1} + \beta_5 lGE_{it} + \beta_6 TO_{it}$$
$$+ \beta_7 lGFCF_{it} + \varepsilon_{it} \tag{2}$$

### 2.2. Data and Variable Description

To check the impact of education and religious freedom on prosperity, 45 lower, middle, and high-income countries are considered. The list of these countries is provided in Appendix A. Religious freedom is based on the indicators that broadly rate the extent of the freedom of religion in society, including the right to practice and choose one's religion, to proselytize peacefully, and to change religions. Similarly, freedom of education is proxied by the freedom of academics, and it is also taken for the period of 2008–2018. The current study employs this time span for two reasons. The first is data availability. The freedom indices data are available from the year 2008. Second, there was a COVID-19 breakdown at the end of 2019. As a result of COVID-19, the current study has been extended until 2018.

The index range for both religion and education is from 0 to 10. The highest value represents the greatest freedom. These data are taken from the human freedom report developed by the Cato Institute and the Fraser Institute. In this study, the Human Development Index (HDI) is used as a dependent variable that determines the level of prosperity. Higher levels of HDI determine a greater level of prosperity. The data of the HDI are obtained from the human development reports of the United Nations from 2008 to 2018 (https://hdr.undp.org/data-center/human-development-index accessed on 1 December 2022). This study uses gross fixed capital formation and trade openness as control variables. The data of these variables are obtained from the World Bank (https://databank.worldbank.org/source/world-development-indicators accessed on 15 December 2022). This study uses government effectiveness to capture the role of governments in prosperity. The data of government effectiveness are from the *Worldwide Governance Indicators* of the *World Bank* (https://info.worldbank.org/governance/wgi/ accessed on 1 December 2022).

### 2.3. Descriptive Statistics

Table 1 provides a descriptive analysis of the dataset. Overall, there are 45 countries in our sample data including higher-income, middle-income, and lower-middle-income countries for 2008 to 2018. The standard deviation of the log of the HDI indicates that the variation within the individual countries during the period under study is considerably lower than variation between the countries. Similarly, the variation in the cases of religious freedom and academic freedom is also low within the countries. This indicates that countries have different levels of HDI and religious freedom but the progress in terms of religious freedom and HDI has been slow during the period of analysis.

**Table 1.** Descriptive Statistics.

| Variable | | Mean | Std. Dev. | Min | Max | Observations |
|---|---|---|---|---|---|---|
| lhdi | overall | −0.23 | 0.14 | −0.66 | −0.04 | N = 494 |
| | between | | 0.14 | −0.62 | −0.06 | n = 45 |
| | within | | 0.02 | −0.32 | −0.13 | T bar = 10.98 |
| lrelig | overall | 2.11 | 0.24 | 1.42 | 2.30 | N = 494 |
| | between | | 0.23 | 1.51 | 2.30 | n = 45 |
| | within | | 0.05 | 1.89 | 2.39 | T bar = 10.98 |
| lacadfe | overall | 2.02 | 0.33 | 0.58 | 2.29 | N = 494 |
| | between | | 0.31 | 1.12 | 2.27 | n = 45 |
| | within | | 0.12 | 1.34 | 2.58 | T bar = 10.98 |
| lgfcf | overall | 25.00 | 1.35 | 21.42 | 27.78 | N = 494 |
| | between | | 1.35 | 21.83 | 27.70 | n = 44 |
| | within | | 0.18 | 24.35 | 25.77 | T bar = 10.34 |
| lto | overall | 16.58 | 1.34 | 12.35 | 20.30 | N = 494 |
| | between | | 1.34 | 12.56 | 20.18 | n = 44 |
| | within | | 0.14 | 16.24 | 16.91 | Tbar = 10.34 |
| lge | overall | −0.33 | 1.24 | −6.53 | 0.81 | N = 494 |
| | between | | 1.55 | −4.72 | 0.74 | n = 44 |
| | within | | 0.39 | −3.84 | 1.32 | T bar = 10.34 |

The rest of the variables, including the log of gross fixed capital formation (lgfcf), the log of trade openness (lto), and the log of government effectiveness (lge), have considerably higher variations across countries and within countries. Figure 1 shows that the HDI is significantly high in high-income countries; this also indicates a close link between income and prosperity. Although the HDI in lower-middle-income countries is considerably lower, it has improved over the years. Moreover, within the group of lower-middle-income countries there is a relatively large variation across countries.

Figure 2 shows the trends in academic freedom in different country by income level. High-income countries tend to have higher academic freedom while on average the level of academic freedom remains more or less the same in lower-middle-income and middle-income countries between 2008 and 2014. After 2018, the index of academic freedom declined in both lower-middle-income and middle-income countries but academic freedom declined more in middle-income countries.

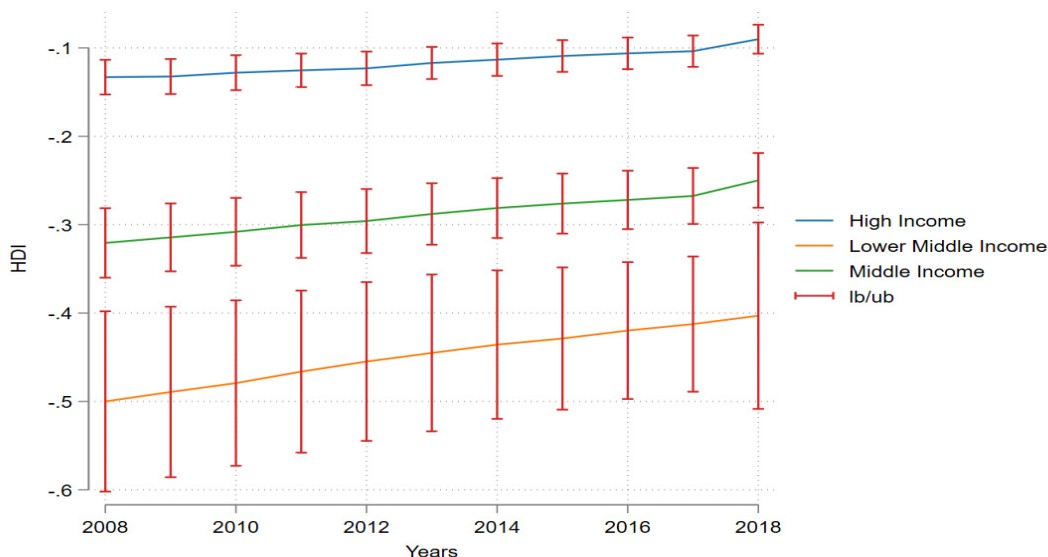

**Figure 1.** Trends in HDI by Income Level.

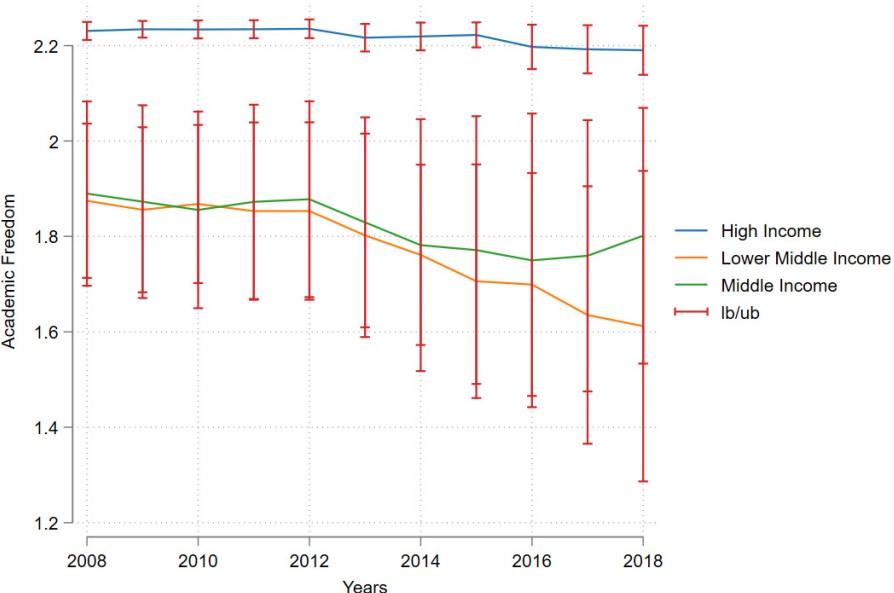

**Figure 2.** Trends in Academic Freedom by Income Level.

Figure 3 shows the trends in religious freedom in different countries by income level. On average, the religious freedom index is highest in high-income countries and there is not much variation between years. However, the lower-middle-income countries have relatively low levels of religious freedom compared to the high-income countries and middle-income countries. Moreover, the status of religious freedom has deteriorated over recent years. Overall, countries have on average improved in terms of the human development index in recent years. However, in terms of religious freedom and academic freedom the lower-middle-income countries have experienced a decline in recent years. Also, in the case of lower-middle-income countries there are huge variations in the HDI, academic freedom, and religious freedom indices. Particularly in the cases of some lower-middle-income countries, there are higher academic and religious freedoms than in middle-income countries.

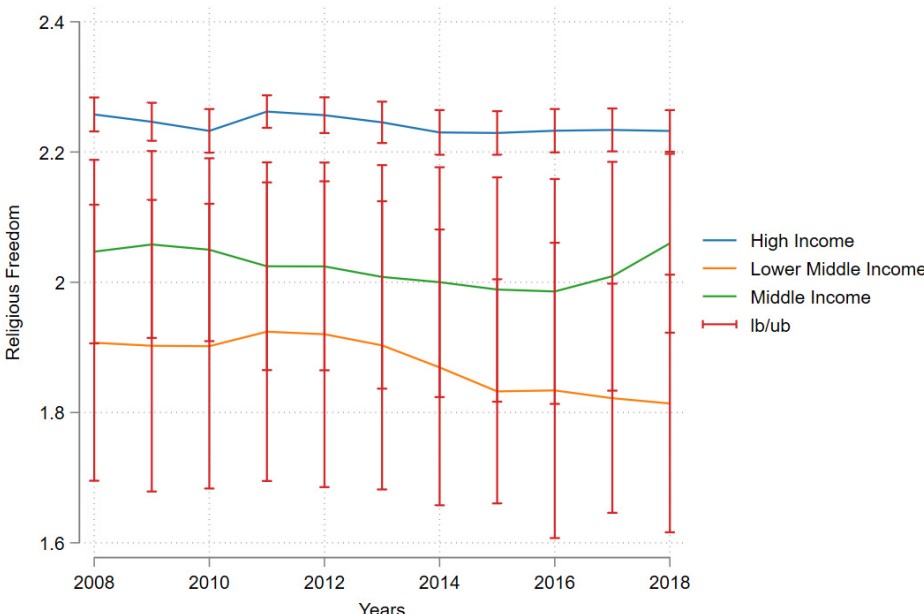

**Figure 3.** Trends in Religious Freedom by Income Level.

Figure 4 shows the relationship between academic freedom and the human development index. Although there is a positive relationship between academic freedom and the human development index, most of the developed countries have higher levels of academic freedom as well as the human development index compared to middle- and lower-middle-income countries. Some of the countries such Azerbaijan (AZE), Thailand (THA), and Malaysia (MYS) are exceptions in the sense that despite low academic freedom their human development is considerably higher than many other countries that have similar or even higher levels of academic freedom. On the other hand, there are also some countries such as Pakistan (PAK), Bangladesh (BAN), and India (IND) that have a considerably lower human development index despite their relatively higher levels of academic freedom.

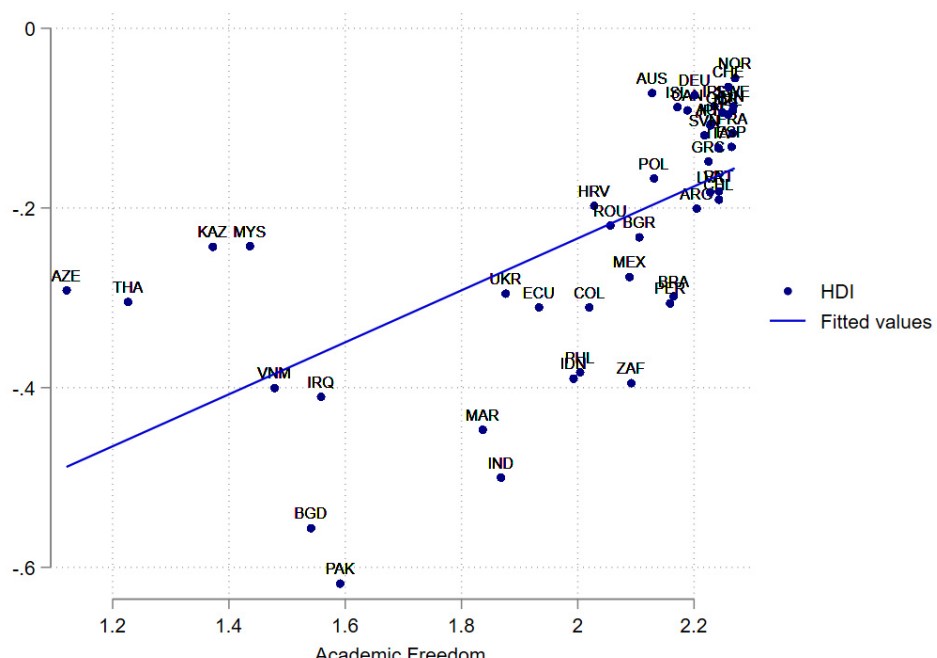

**Figure 4.** The HDI and Academic Freedom.

Figure 5 shows the relationship between religious freedom and the human development index. Most of the developed countries have higher levels of religious freedom as well as the human development index compared to middle- and lower-middle-income countries. Pakistan has the lowest level of religious freedom and human development index within the sample countries. Some of the countries such India (IND) and Bangladesh have a higher level of religious freedom compared to many lower-middle-income countries but their human development index is relatively low.

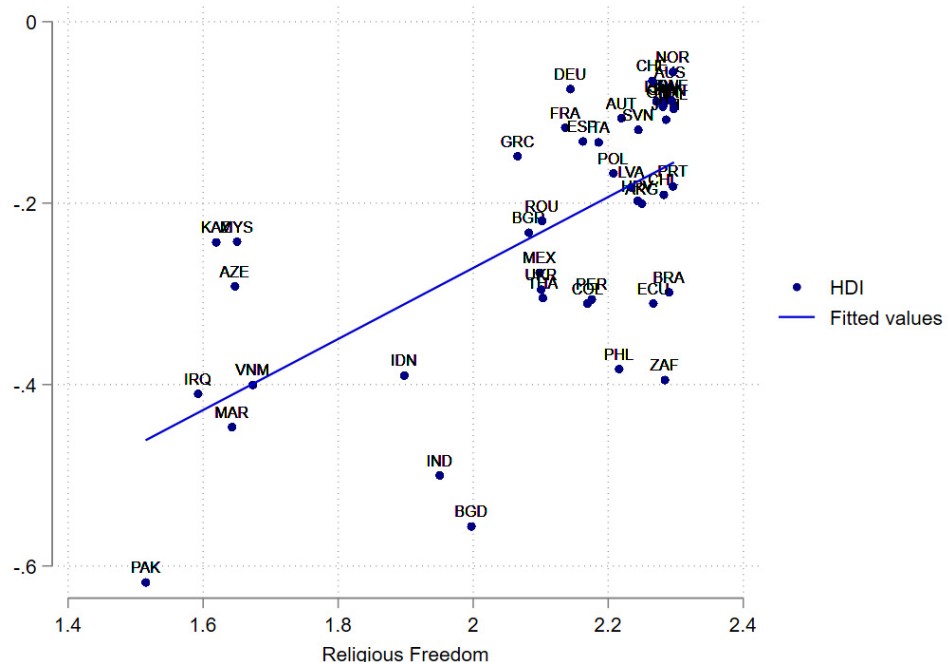

**Figure 5.** The HDI and Religious Freedom.

Overall, academic freedom and religious freedom have a positive relationship with the human development index. However, some of the countries tend to have a lower HDI despite reasonably higher academic and religious freedoms. Firstly, academic freedoms give rise to productivity and provide opportunities for increasing the skills and productivity of individuals. Mostly, high-income countries have academic freedom and higher levels of productivity that are later translated into higher labor force participation and economic growth. On the other hand, religious freedom also affects economic prosperity simply because religious freedom promotes a pluralism that is linked to economic expansion. For instance, between 2008 and 2012, each of the world's top 12 religiously diverse nations outperformed average global economic growth.[1]

## 3. Results and Discussion

Table 2 gives the results of the two-step System GMM model's estimations with robust standard errors in parentheses. The lagged dependent variables are treated as predetermined variables and academic freedom and religious freedom along with other control variables are treated as endogenous variables, while the years' dummies are exogenous instruments. The estimates of the two-step system GMM model show that for all the specifications the lagged human development index is positive and significant, indicating that past levels of the HDI have a large positive impact on the current HDI. All the models satisfy the tests of the validity of the instruments. The difference-in-Hansen (IV) test (*p*-value = 0.244) shows that the exogenous instruments used in the model are valid and fulfill the condition of orthogonality. Also, the *p*-value of AR(2) is more than 5 percent indicating there is no second order autocorrelation. Furthermore, the Hansen-J statistics and difference in Hassen-J statistics indicate the validity of the institution.

The results show that both academic freedom and religious freedom are positively and statistically significantly associated with the human development index. These results imply that that restrictions on religion or religious activities by the government tend to exert a negative force on human development. The key channel through which religious freedom/restrictions affect human development is social hostilities involving religion and religious practices. The restrictions over religion can be in the form of favoritism towards a particular sect or non-believers. Governments should protect all religions and impose only impartial limitations. The results of religious and educational freedoms and their impacts on prosperity are supported by the previous study of Makrevska Disoska and Kocevska (2019).

**Table 2.** System GMM Estimations.

|  | (1) | (2) | (3) |
|---|---|---|---|
|  | lhdi | lhdi | lhdi |
| L.hdi | 1.361 *** | 1.450 *** | 1.633 *** |
|  | (0.0466) | (0.0392) | (0.0436) |
| lacadfree | 0.00328 | 0.0133 | 0.0389 *** |
|  | (0.00711) | (0.00729) | (0.00815) |
| L.lacadfree | 0.0305 *** | 0.0387 *** | 0.757 *** |
|  | (0.00421) | (0.00656) | (0.148) |
| lrelig | 0.0284 *** | 0.0325 ** | 0.0671 *** |
|  | (0.00775) | (0.00996) | (0.0156) |
| L.lrelig | 0.0433 * | 0.0758 ** | 0.864 *** |
|  | (0.0173) | (0.0220) | (0.165) |
| lge | 0.00328 | −0.0794 * | 0.0879 * |
|  | (0.00178) | (0.0390) | (0.0472) |
| lgfcf | −0.00637 | 0.00925 * | 0.0255 ** |
|  | (0.00318) | (0.00358) | (0.00788) |
| lto | 0.00365 ** | 0.00271 | 0.00331 |
|  | (0.00111) | (0.00298) | (0.00309) |
| c.lge#c.lgfcf |  | 0.00332 * | 0.00365 * |
|  |  | (0.00152) | (0.00126) |
| cL.lacadfree#cL.lrelig |  |  | 0.378 *** |
|  |  |  | (0.0710) |
| N | 249 | 249 | 249 |
| AR(1) $p$ | 0.017 | 0.013 | 0.038 |
| AR (2) $p$ | 0.663 | 0.408 | 0.841 |
| Hansen $p$ | 0.674 | 0.529 | 0.395 |
| Diff-Hansen (GMM) | 0.409 | 0.482 | 0.430 |
| Diff-Hansen (IV) | 0.98 | 0.551 | 0.286 |

Standard Errors in parentheses *** $p < 0.01$, ** $p < 0.05$, * $p < 0.1$. Source: Author's estimations.

Academic freedom is also important for human development, simply because it opens employment opportunities. It expands the choices of people in terms of developing their skills according to their desires and beliefs. The lack of freedom of education, particularly for the poor and marginalized sections of the population, not only decreases their income but it also affects their lifetime health outcomes and productivity. The fully specified model (3) shows that the lagged impact of both religious and academic freedom has a higher impact on the human development index than the current level of both the variables. This indicates that freedom of religion and education has a long-term impact on human development.

Government effectiveness, gross fixed capital formation, and trade openness also have the expected signs. They have a positive impact on the human development index, but the impact of trade openness on the human development index is insignificant. Interestingly, the interaction term between academic and religious freedoms is also positive and statistically significant, which indicates that their combination affects further increases in the human development index (see model 2).The interaction term between government effectiveness and gross fixed capital formation is positive and significant, indicating that capital investment has a higher positive effect on the human development index in cases of higher government effectiveness (see model 3). Previous research by Lee et al. (2019) and Joharee (2018) supports these findings of governance and its influence on prosperity.

## 4. Conclusions

This research looked at the effect of religious and educational freedom on prosperity. The system GMM model is used for this purpose, including data from 45 poor, medium-, and high-income countries from 2009 to 2018. According to the findings, religious and educational freedoms are both positively and statistically substantially connected with success. The findings show that the lag effect of religious and academic freedom has a greater influence on prosperity than the present levels of both variables. Interestingly, the interaction term between academic and religious freedoms is both positive and statistically significant, implying that their combined effect raises the human development index even more. Furthermore, the interaction term between government effectiveness and gross fixed capital formation is included, with a positive and substantial influence, demonstrating that capital investment has a greater positive effect on the human development index when government effectiveness is higher. This research used gross fixed capital creation and trade openness as control variables, and although these factors have a favorable influence on prosperity, their impact on the human development index is negligible.

Despite widespread agreement that access to education and freedom of religion are fundamental human rights, political leaders at all levels are responsible for establishing laws and procedures that protect and promote religious liberty, in conformity with previously agreed global standards. Governments should go much farther by declaring the free and equal access to education a fundamental human right.

Since the world is becoming smaller, owing to an increased interdependence between and among governments and their peoples, protecting the rights of all students to an education that is free from discrimination on the basis of religion or ethnicity is of paramount importance. Therefore, if we want people of various religions to be able to coexist peacefully, we need to ensure that they receive an education that welcomes and celebrates diversity. As daunting and all-encompassing as it may seem, educational leaders, lawmakers, and policymakers must work to ensure that all students have access to a quality education and religious freedom, as today's students will one day become tomorrow's leaders and work to make the world a better place for everyone.

Due to the lack of freedom indices, the current study has a data limitation in that it only analyzes data from 2008 to 2018. However, the extended time period will be used in future study along with taking the COVID-19 influence into account.

**Author Contributions:** Conceptualization, K.A.K. and D.A.; Investigation, K.A.K. and Y.W.; Resources, K.A.K.; Writing—original draft, K.A.K.; Writing—review & editing, D.A. and Y.W. All authors have read and agreed to the published version of the manuscript.

**Funding:** This research received no external funding.

**Institutional Review Board Statement:** Not applicable.

**Informed Consent Statement:** Informed consent was obtained from all subjects involved in the study.

**Data Availability Statement:** Not applicable.

**Conflicts of Interest:** The authors declare no conflict of interest.

## Appendix A

| | List of Countries | | |
|---|---|---|---|
| **No.** | **Name of Countries** | **No.** | **Name of Countries** |
| 1 | Argentina | 24 | Kazakhstan |
| 2 | Australia | 25 | Latvia |
| 3 | Austria | 26 | Malaysia |
| 4 | Azerbaijan | 27 | Mexico |
| 5 | Bangladesh | 28 | Morocco |
| 6 | Brazil | 29 | New Zealand |
| 7 | Bulgaria | 30 | Norway |
| 8 | Canada | 31 | Pakistan |
| 9 | Chile | 32 | Peru |
| 10 | Colombia | 33 | Philippines |
| 11 | Croatia | 34 | Poland |
| 12 | Ecuador | 35 | Portugal |
| 13 | Finland | 36 | Romania |
| 14 | France | 37 | Slovenia |
| 15 | Germany | 38 | South Africa |
| 16 | Greece | 39 | Spain |
| 17 | Iceland | 40 | Sweden |
| 18 | India | 41 | Switzerland |
| 19 | Indonesia | 42 | Thailand |
| 20 | Iraq | 43 | Ukraine |
| 21 | Ireland | 44 | United Kingdom |
| 22 | Italy | 45 | Vietnam |
| 23 | Japan | | |

## Note

[1] The link between economic and religious freedoms (20 May 2022), World Economic Forum. Retrieved 22 September 2022, from https://www.weforum.org/agenda/2014/12/the-link-between-economic-and-religious-freedoms/.

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
