# Peer review of "Do Countries Need Religious and Educational Freedoms to Achieve Prosperity?"

_religions, doi:10.3390/rel14010112_

Round 1

Reviewer 1 Report

The paper is well-structured and presents good bibliographical references. It is very original and shows a good competence in the use of different research techniques. 

Author Response

It is a great privilege to receive comments from you. 

The revised manuscript is further improved. I appreciate your consideration. 

Reviewer 2 Report

Your article is interesting. Please, note:

l. 66-67: your sentence is not clear to me

l. 200: uses?

l. 204:   ,

l. 209: that, that?

l. 358: pp.?

l. 359: &Bover     & Bover

l. 361: &Griera    & Griera

l. 375: pp.?

l. 388: &ShapkovaKocevska   & Shapkova Kocevska

l. 390: &Karabenick   & Karabenick

l. 396: 629693   629-693

l. 399: &Sabel   & Sabel

Author Response

It is a great privilege to receive comments from honourable reviewers. All the comments have been incorporated in the revised manuscript

Cooment

66-67: your sentence is not clear to me

Response: The sentence is revised 

Comment

l. 200: uses?

Response: The word is rephrased with used

Comment

l. 204: ,

Response: The sentence is improved

Comment

l. 209: that, that?

Response: The correction has been made. 

Comment

l. 358: pp.?

Response:  Page number is added

Comment

l. 359: &Bover     & Bover

Response: Reference is corrected

Comment

l. 361: &Griera    & Griera

Response: Corrected

Comment

l. 375: pp.?

Response: Reference is updated

Comment

l. 388: &ShapkovaKocevska   & Shapkova Kocevska

Response: Corrected

Comment

l. 390: &Karabenick   & Karabenick

Response: Corrected

Comment

l. 396: 629693   629-693

Response: Corrected

Comment

l. 399: &Sabel   & Sabel

Response: Corrected

Reviewer 3 Report

The first lines of the article immediately raise questions of plagiarism. The first reference (Thomas, 2010) opens with the lines, "Around the world - from the southern United States to the MIddle East - religion is on the rise. It is growing in countries with a wide variety of religious traditions and levels of economic development, suggesting that neither poverty nor social exclusion is solely responsible."

Compare this manuscript: "Religion is growing all throughout the world, from the southern United States to the Middle East (Thomas, 2010). It is expanding in countries with diverse religious traditions and degrees of economic development, suggesting that neither poverty nor social isolation is to blame (Iannaccone, 1998)." The second reference (Iannacone, 1998) is clearly misplaced.

The article on line 40 referred to as (Kochhar, 2014) has nothing to do hostility and nowhere discusses 74% hostility.

In any case, both the above lines are suspect: first, because religion is surely on the wane in parts of the world such as Europe and secondly because it is far-fetched to believe that nearly 3/4 of people live in areas of high hostility.

On line 50 is a reference to (Grim and Finke, 2007), which is not listed in the reference list.

A third example of plagiarism is lines 34-35, which read, "During the early years of the twenty-first century, a wave of religious hostility swept the world. The terrorist acts of September 11, 2001, which triggered conflicts in Afghanistan and Iraq, demonstrated that even a small number of individuals motivated by religious extremism might spark wars and severe economic disruptions (Mockaitis, 2008)."

Compare to (Grim et al, 2014, 3): "A wave of religious hostilities has swept the globe during the early years of the 21st century. The terrorist attacks of September 11, 2001, leading to the wars in Afghanistan and Iraq, made clear that even a few people motivated by religious extremism can trigger wars and major economic disruptions."

Unfortunately, I cannot proceed with reviewing the paper because I cannot take seriously the merits of the core arguments of the paper because the introduction is not only wrong, it is plagiarized.

Round 2

Reviewer 3 Report

This looks better. Thank you for removing the plagiarized sections.

My main recommendation is to justify the time period 2009-2018 for your research. This is an unusual time, because it starts after the Global Financial Crisis of 2008-2009, which should skew the data in one direction or another. If you have chosen the data simply because it is the most convenient, then says so. If there is a reason that earlier time periods (or a broader time period is not included), then make that also clear.

Author Response

It is a great privilege to receive comments from the honourable reviewer. 

- The comment is well taken, and in the revised manuscript, the justification of the time period 2008-2018 is added in the manuscript. Please see lines 182-185

-The data of freedom indices are not available before 2008. Due to data limitations, we have not considered the longer time period or prior to 2008. However, this limitation is mentioned at the end of the conclusion, along with future studies. Please see lines 344-347.